# On the Prevalence and Roles of Proteins Undergoing Liquid–Liquid Phase Separation in the Biogenesis of PML-Bodies

**DOI:** 10.3390/biom13121805

**Published:** 2023-12-18

**Authors:** Sergey A. Silonov, Yakov I. Mokin, Eugene M. Nedelyaev, Eugene Y. Smirnov, Irina M. Kuznetsova, Konstantin K. Turoverov, Vladimir N. Uversky, Alexander V. Fonin

**Affiliations:** 1Laboratory of Structural Dynamics, Stability and Folding of Proteins, Institute of Cytology, Russian Academy of Sciences, St. Petersburg 194064, Russia; silonovsa@incras.ru (S.A.S.); mokinyakov@mail.ru (Y.I.M.); nedelyaev99@mail.ru (E.M.N.); e.smirnov@incras.ru (E.Y.S.); imk@incras.ru (I.M.K.); kkt@incras.ru (K.K.T.); 2Department of Molecular Medicine and USF Health Byrd Alzheimer’s Research Institute, Morsani College of Medicine, University of South Florida, Tampa, FL 33612, USA; vuversky@usf.edu

**Keywords:** PML bodies, intrinsically disordered proteins, intrinsically disordered regions, liquid–liquid phase separation, membrane-less organelles, protein–protein interactions, post-translational modifications, SUMOylation

## Abstract

The formation and function of membrane-less organelles (MLOs) is one of the main driving forces in the molecular life of the cell. These processes are based on the separation of biopolymers into phases regulated by multiple specific and nonspecific inter- and intramolecular interactions. Among the realm of MLOs, a special place is taken by the promyelocytic leukemia nuclear bodies (PML-NBs or PML bodies), which are the intranuclear compartments involved in the regulation of cellular metabolism, transcription, the maintenance of genome stability, responses to viral infection, apoptosis, and tumor suppression. According to the accepted models, specific interactions, such as SUMO/SIM, the formation of disulfide bonds, etc., play a decisive role in the biogenesis of PML bodies. In this work, a number of bioinformatics approaches were used to study proteins found in the proteome of PML bodies for their tendency for spontaneous liquid–liquid phase separation (LLPS), which is usually caused by weak nonspecific interactions. A total of 205 proteins found in PML bodies have been identified. It has been suggested that UBC9, P53, HIPK2, and SUMO1 can be considered as the scaffold proteins of PML bodies. It was shown that more than half of the proteins in the analyzed proteome are capable of spontaneous LLPS, with 85% of the analyzed proteins being intrinsically disordered proteins (IDPs) and the remaining 15% being proteins with intrinsically disordered protein regions (IDPRs). About 44% of all proteins analyzed in this study contain SUMO binding sites and can potentially be SUMOylated. These data suggest that weak nonspecific interactions play a significantly larger role in the formation and biogenesis of PML bodies than previously expected.

## 1. Introduction

Fundamental changes in ideas about the spatiotemporal organization of intracellular space and the organization and control of biochemical processes in the cell occurred in the 2010s [1,2,3]. It has become clear that the formation and function of biomolecular condensates—membrane-less organelles (MLOs)—is one of the main driving forces in the molecular life of a cell. These processes are based on the separation of biopolymers, such as primarily intrinsically disordered proteins (IDPs) and RNA, into phases regulated by multiple specific/nonspecific inter- and intramolecular interactions [4,5]. IDPs and proteins containing intrinsically disordered protein regions (IDPRs) are known to play a decisive role in liquid–liquid phase separation (LLPS), leading to the formation of MLOs [6]. This is because, as a rule, IDPs/IDPRs contain blocks of the same type of amino acid residues, multiple weak nonspecific interactions (electrostatic, π–π, cation–π) between which, under conditions of macromolecular crowding, can help to overcome the potential energy barrier required for the transition of IDPs to the condensed liquid-droplet phase [7,8].

MLOs include nuclear PML bodies (also known as ND10)—compartments involved in the regulation of cellular metabolism, transcription, the maintenance of genome stability, responses to viral infection, apoptosis, and tumor suppression [9,10]. PML bodies are subnuclear spherical and toroidal structures with a diameter of 0.1–2 μm [11]. They are present in most types of mammalian cells and tissues. Depending on the cell type, cell cycle phase, and differentiation stage, there may be about 5–30 PML bodies per cell [12]. Based on an immunofluorescence analysis, PML bodies exist as nuclear dot-shaped spherical structures residing in the interchromatin nuclear space [13]. As a rule, large PML bodies have a toroidal structure formed of a ≈100 nm thick shell at the periphery and a more mobile internal core [14]. According to the accepted models, canonical PML bodies are protein assemblages that do not contain nucleic acids [13]. However, specific types of PML bodies are known that are capable of incorporating DNA/RNA into their composition [9]:-Viral DNA-containing PML bodies;-Giant PML bodies containing satellite DNA;-Alternative lengthening of telomeres (ALT)-associated PML bodies.

The major scaffolding protein of PML bodies is promyelocytic leukemia protein (PML) [15]. Due to alternative splicing, the PML protein is represented by seven main isoforms transcribed from one gene: six with nuclear localization (PML-I to PML-VI) due to the presence of the nuclear localization signal (NLS) in exon 6, and one with cytoplasmic localization (PML-VII). The sequences encoded by the first four exons of the *PML* gene (418 amino acids with the conserved RBCC motif consisting of a RING finger domain (R), followed by two cysteine/histidine-rich B-box domains (B) and an α-helical coiled-coil domain (CC)) are common to all isoforms and contain structurally conserved domains: proline-rich (1–45 a.a.), RING (45–105 a.a.), B1-box (124–166 a.a.), B2-box (184–229 a.a.), and coiled-coil (229–323 a.a.) [16] (Figure 1).

According to modern understanding, specific interactions play a decisive role in the formation and biogenesis of PML bodies [17]. It is assumed that the oxidative formation of disulfide bonds between PML molecules and non-covalent interactions in the region of the ordered RBCC motif of PML trigger the oligomerization of non-SUMOylated PML proteins, which subsequently undergo UBC9-mediated (poly-)SUMOylation and recruit PML-body client proteins via SUMO–SIM interactions [17,18].

We have previously shown that a certain role in the formation of PML bodies is played by nonspecific intra- and intermolecular interactions that can lead to the spontaneous phase separation of at least the disordered C-terminal domains of the PML-II and PML-V isoforms [19]. In this regard, we decided to analyze the proteins included in the proteome of PML bodies for their tendency to spontaneously separate into phases, and, consequently, for their ability to be nonspecifically included in PML bodies.

## 2. Materials and Methods

### 2.1. Study Design

The identification of the proteins included in the PML-body proteome was carried out using two approaches:(i).An analysis of the PML-body proteome and the PML protein interactome in the BIOGRID (v.4.4) databases https://thebiogrid.org/ (accessed on 10 October 2023), and resources https://amigo.geneontology.org/amigo/term/GO:0016605 (accessed on 10 October 2023) (GO:0016605) (PML NB/ND10) and SL-0465 (https://www.uniprot.org/locations/SL-0465 (accessed on 10 October 2023);(ii).An analysis of data from the literature.

For further analysis, 205 proteins were selected that could be localized in PML bodies according to fluorescent microscopy or electron microscopy data. Figure 2 schematically represents the design of our study.

### 2.2. Intrinsic Disorder Prediction

An analysis of the intrinsic disorder predispositions of the studied proteins was performed using the RIDAO online platform [20]. This platform predicts potential per-residue disorder in proteins based on their amino acid sequences, using predictors based on different algorithms such as PONDR^®^ FIT, PONDR^®^ VLXT, PONDR^®^ VLS2, PONDR^®^ VL3, IUPred-Long, and IUPred-Short [21,22,23,24,25,26]. The PONDR^®^ VSL2 algorithm showed the most correct prediction of disorder for a large data set. Consequently, the PONDR^®^ VSL2 (%), the percentage of predicted intrinsically disordered residues (PPIDR), was employed as an indicator of protein structural disorder. This indicator reflects the percentage of residues in a protein’s amino acid sequence for which the disorder prediction (disorder score) is >0.5 (“probably disordered” according to the algorithm).

Typically, a PPIDR value below 10% indicates a protein with high order, while a PPIDR ranging between 10% and 30% is assigned to proteins characterized as moderately disordered. Proteins with a PPIDR exceeding 30% are classified as highly disordered [27,28]. Also, the mean disorder score (MDS) was computed for each queried protein as a length-normalized sum of the per-residue disorder scores. Proteins were grouped based on their respective MDS values, with those having an MDS of less than 0.15 considered highly ordered; an MDS between 0.15 and 0.5 categorized them as moderately disordered or flexible, and those with a MDS equal to or greater than 0.5 were designated as highly disordered.

The intrinsic disorder predisposition of the analyzed data set was further evaluated via the CH-CDF analysis, the data for which were generated using RIDAO [21]. The analysis integrates outcomes derived from two binary predictors of disorder, namely the cumulative distribution function (CDF) and charge–hydropathy (CH) plot analysis [29,30]. The combination of data generated using these approaches results in the ΔCH-ΔCDF plot, allowing discrimination between flavors of disorder based on the positions of proteins within the quadrants of the ΔCH-ΔCDF phase space [31,32]. In this context, proteins positioned in the top-left quadrant are anticipated to exhibit disorder according to both the CH and CDF predictions, while those in the bottom-left quadrant are projected to be compact or ordered based on CH but disordered according to CDF. Proteins situated in the top-right quadrant are expected to be disordered according to CH but ordered based on the CDF, whereas those in the bottom-right quadrant are predicted to be ordered based on both the CH and CDF predictors [31,32].

### 2.3. LLPS Predisposition Analysis

The FuzDrop [33] and PSPredictor [34] predictors were used for the LLPS analysis of the studied proteins. The proteins with a PSPredcitor score > 0.5 and FuzDrop score > 0.6 were marked as prone to LLPS. If the results of the FuzDrop and PSPredictor analyses disagreed, the analyzed proteins were classified as the “controversial LLPS” group.

The FuzDrop predictor also allows one to predict the ability of the analyzed protein to undergo spontaneous and interaction-induced LLPS. Proteins that spontaneously undergo liquid–liquid phase separation are termed as droplet-driving proteins. Proteins that require additional interactions to form droplets are termed as clients. Proteins with a pLPS ≥ 0.60 likely drive liquid–liquid phase separation. Proteins with droplet-promoting regions, defined as consecutive residues with a pDP ≥ 0.60, will likely serve as droplet clients.

### 2.4. Evaluation of Protein Charge and Hydrophobicity

The charge of the examined proteins at pH 7.0 was determined as the average charge across the protein sequence, considering the charge values of amino acids: D = −1, E = −1, H = 0.5, R = 1, K = 1 [29]. The average hydrophobicity of the investigated proteins was computed using the normalized Kite and Doolittle scale [35].

### 2.5. Determination of Proteins Targeted for SUMOylation

This was performed by visiting the UniProt website (https://www.uniprot.org/, accessed on 10 September 2023) [36] and then searching for mentions of SUMO in the PTM/Processing section of the corresponding UniProt entry.

### 2.6. Aggregation Propensity Prediction

The assessment of the proteins’ propensity for aggregation and the formation of amyloid fibrils was conducted through the AggreScan package [37], enabling the identification of aggregation hot spots—regions of the protein that facilitate its aggregation.

### 2.7. Determination of the Biological Processes and Molecular Functions of Proteins

Gene Ontology (GO) was performed using the Enrichr resource (https://maayanlab.cloud/Enrichr/, accessed on 10 September 2023), and 10 terms with the lowest *p*-values were selected [38]. The *p*-values were computed using the Fisher exact test, a statistical method employed to determine the presence of significant non-random associations between two categorical variables.

## 3. Results and Discussion

### 3.1. Finding Proteins Included in the PML-Body Proteome

PML bodies take part in a large number of intracellular processes [9,10,39,40,41,42,43]. In this regard, the composition of the client proteins of PML bodies varies significantly. In addition, proteins that interact with PML are often included in the PML-body proteome [15,44]. However, PML protein is localized in the cell, not just in PML bodies. It is known that several isoforms of the PML protein can be localized in the cytoplasm [16,45], and the PML bodies themselves pass into the cytoplasm during mitosis [40]. Taken together, this complicates the identification of proteins that are uniquely present in PML bodies.

In 2010, mainly based on an analysis of data from the literature, the so-called PML-body interactome was proposed, combining 166 proteins, most of which are part of the SUMO conjugation pathway [44]. A 2015 paper [15] discussed a network of 120 proteins that are involved in direct physical interactions with PML, most of which are part of PML bodies. This set included proteins whose interaction with PML was confirmed via affinity capture followed by Western blotting (BIOGRID data). Proteins associated with PML, identified using high-throughput methods, were not discussed in this work [15].

An analysis of more than 150 works [44,46,47,48,49,50,51,52,53,54,55,56,57,58,59,60,61,62,63,64,65,66,67,68,69,70,71,72,73,74,75,76,77,78,79,80,81,82,83,84,85,86,87,88,89,90,91,92,93,94,95,96,97,98,99,100,101,102,103,104,105,106,107,108,109,110,111,112,113,114,115,116,117,118,119,120,121,122,123,124,125,126,127,128,129,130,131,132,133,134,135,136,137,138,139,140,141,142,143,144,145,146,147,148,149,150,151,152,153,154,155,156,157,158,159,160,161,162,163,164,165,166,167,168,169,170,171,172,173,174,175,176,177,178,179,180,181,182,183,184,185,186,187,188,189,190,191,192,193,194,195,196,197,198,199,200,201,202,203,204,205,206,207,208,209,210,211,212,213,214,215,216,217,218] allowed us to identify 205 proteins that can be localized in PML bodies according to fluorescent microscopy or electron microscopy data. These proteins were selected for further analysis.

Also, we estimated the ratio of proteins capable of directly interacting with PML protein in a selected array. For this purpose, we analyzed the PML interactome from the BIOGRID database (v.4.4) [219], selecting proteins involved in the biogenesis of these membrane-less organelles. From the 707 proteins included in the network of molecular interactions of the PML protein, we selected 267 proteins for which an interaction with PML was shown at least twice. Using information obtained from the resources https://amigo.geneontology.org/amigo/term/GO:0016605 (accessed on 10 October 2023) (GO:0016605) (PML NB/ND10) and SL-0465 (https://www.uniprot.org/locations/SL-0465 (accessed on 10 October 2023)), for further analysis, we selected 104 proteins that are potentially included in human PML bodies.

A comparison of the obtained data showed that about of quarter of the proteins selected via the literature analysis (52 of 205) were not identified in analyzed bioinformatics databases as PML-binding proteins (Figure 3).

### 3.2. PML Bodies’ Scaffold Proteins

PML bodies have complex a structure, formed of scaffold proteins that are constantly present in PML bodies and client proteins that are temporarily localized in PML bodies. It is suggested that the scaffold proteins drive the PML bodies’ formation.

It is known that the scaffold proteins of PML bodies include PML, death domain-associated protein (DAXX), cyclic AMP-responsive element-binding protein (CREB)-binding protein (CREBBP), and speckled 100 kDa (SP100, also known as nuclear dot-associated Sp100 protein or nuclear autoantigen Sp-100) [220]. These proteins were also included in the analyzed array, selected based on data from the literature.

We analyzed the interactomes of these four proteins and found that, according to the BIOGRID database, the proteins ubiquitin carrier protein 9 (UBC9, also known as SUMO-conjugating enzyme UBC9), small ubiquitin-related modifier 1 (SUMO1), p53, and homeodomain-interacting protein kinase 2 (HIPK2) are included in all four interactomes analyzed (Figure 4).

The analysis of the literature data indicates that these proteins are one way or another included in the biogenesis of PML bodies (UBC9 [222], SUMO1 [223], HIPK2 [147], and p53 [224,225,226]). As is known, PML bodies are a platform for the SUMOylation and ubiquitination of various proteins [227]. The SUMO-conjugating enzyme, UBC9, accepts ubiquitin-like proteins SUMO1, SUMO2, SUMO3, SUMO4, and SUMO1 pseudogene 1 (SUMO1P1/SUMO5) from the ubiquitin-like 1-activating enzyme E1A–ubiquitin-like 1-activating enzyme E1B (UBLE1A-UBLE1B) complex and catalyzes their covalent attachment to other proteins using E3 ligases, such as Ran-binding protein 2 (RANBP2, also known as E3 SUMO-protein ligase RanBP2 or nucleoporin Nup358), chromobox protein homolog 4 (CBX4, also known as E3 SUMO-protein ligase CBX4 or Polycomb 2 homolog, Pc2), and zinc finger protein 451 (ZNF451, also known as E3 SUMO-protein ligase ZNF451) [227,228]. UBC9 can catalyze the formation of poly-SUMO chains. It is known that the SUMOylation of proteins included in PML bodies occurs in a UBC9-dependent manner [229]. The conjugation of SUMO substrates often requires an E3 ligase, which provides substrate specificity by simultaneously binding UBC9 and the substrate. E3 SUMO ligases typically use the RING domain to interact with UBC9. The PML protein is considered a possible SUMO ligase [220]. The antiparallel PML dimer model has been proposed to potentially support the accessibility of the PML B-box1 domain for UBC9 binding [229].

PML bodies are also a platform that provides post-translational modifications (PTMs) of the tumor suppressor p53, allowing this protein to take part in the regulation of a number of intracellular processes, in particular, senescence [230]. It is worth noting that one of the p53 regulatory proteins that serves as a part of PML bodies is the ubiquitin carboxyl-terminal hydrolase 7 (USP7) protein [231]. Homeodomain-interacting protein kinase 2 (HIPK2) promotes the p300-mediated acetylation of p53 at K382 and influences the selective transactivation of proapoptotic p53 target genes. To cause the HIPK2-mediated induction of pro-apoptotic p53 genes, p53 must be modified by both S46 phosphorylation and K382 acetylation [232]. It is suggested that SP100 acts as a coactivator of the HIPK2-mediated activation of p53, and thus has an essential role in p53-dependent gene expression [233]. p300 (CREBBP homolog) and CBP (cyclic AMP response element-binding protein-binding protein) directly bind to p53 and acetylate several lysine residues located in the C-terminal domain of this protein: K370, K372, K373, K381, K382, and K386. Furthermore, four additional lysine residues, K101, K139, K164, and K305, can be acetylated by p300/CBP. It has been suggested that the acetylation of these sites enhances the DNA-binding ability of p53 and promotes its transcription, leading to growth arrest or apoptosis [232].

Taken together, these data suggest that UBC9, SUMO1, p53, and HIPK2 can be included in the set of PML scaffolding proteins.

### 3.3. Proteome Analysis of PML Bodies

We analyzed 205 proteins that were identified as proteins of the PML-body proteome for their predisposition for intrinsic disorder and tendency. The results of this analysis are summarized in Figure 5.

The integrative disorder analysis of the proteins that can be localized in PML bodies revealed an exceptionally high prevalence of intrinsic disorder. Approximately 14.6% of these proteins (Figure 5A) were predicted to be moderately disordered (dark-pink segment). An additional 17.1% of proteins were anticipated to be moderately disordered based on their MDS values (segment with light-pink color), while the majority (68.3%) of proteins in the PML-body proteome were expected to be mostly disordered (located in the red-colored block). Importantly, a significant proportion of these mostly disordered proteins (131 out of 140 or 93.6%) exhibited PPIDR values ≥50% and MDS values ≥0.5, indicating their highly disordered nature. This global disorder content in the PML-body proteome dramatically exceeds that of the human proteome, where the moderately and highly disordered proteins account for 51.7% and 39.8%, respectively [234].

Additional evidence supporting the remarkably high disorder status of human proteins within the PML-body proteome was obtained by analyzing the combined outcomes of two binary disorder predictors: cumulative distribution function (CDF) and charge–hydropathy (CH) plot analysis, as depicted in Figure 5B. Figure 5B highlights that only 34.3% of these proteins (in contrast to 59.1% in the entire human proteome [234]) refer to the Q1 quadrant (bottom-right corner) that includes proteins predicted to be ordered. Meanwhile, 65.7% of these proteins are located outside the Q1 quadrant, suggesting their elevated disorder levels. A total of 42.9% of the studied proteins are located within the Q2 quadrant (bottom-left corner), corresponding to molten globular or hybrid proteins (in comparison to 21.5% in the human proteome [234]). Additionally, 21.5% of the proteins analyzed in this study are found in the Q3 quadrant (top-left corner), indicating a prediction of high disorder levels with both predictors. In quadrant Q4 (top-right corner), only 1.5% of proteins are located (compared to 3.1% in the human proteome [234]).

This hypothesis was supported by the analysis of the predisposition of human proteins from the PML-body proteome to undergo spontaneous LLPS.

The conducted analysis revealed (Figure 6, Appendix A) that there are practically no globular proteins in the analyzed data set, which differs significantly from the human proteome (the proportion of IDPs in which is about 40% [234]). This is also consistent with the data that 52% of the analyzed proteins in the PML-body proteome are highly likely to be capable of spontaneous phase separation (Figure 6E). Furthermore, according to the FuzDrop analysis, the number of proteins that are “LLPS drivers”, i.e., proteins capable of spontaneous phase separation in the proteome of PML bodies, is almost twice as large as the number of client proteins, i.e., proteins capable of LLPS upon interaction with some partners (Figure 6C, Appendix A). Scaffold proteins of PML bodies (including the main PML-body isoforms) also are presented as predominantly LLPS-drivers and IDPs (Table 1 and Table 2).

However, only 44% of all proteins studied contain SUMO binding sites and can potentially be SUMOylated (Figure 6B).

The data obtained indicate that the proportion of proteins in the proteome of PML bodies that are capable of nonspecific inclusion in these structures is at least no less than the proportion of proteins included in these MLOs due to SUMO/SIM interactions.

Proteins prone to spontaneous LLPS are mainly represented by transcription factors, regulatory elements, DNA-damage response (DDR) proteins, and proteins involved in the alternative lengthening of telomeres (Figure 7). About 10% of LLPS-prone proteins are DNA-binding proteins.

This abundance of DNA-binding proteins in the proteome of PML bodies suggests that DNA is somehow involved in the biogenesis of these MLOs. This assumption is also supported by the fact that more than half of the proteins studied are positively charged, which may indicate their nonspecific interaction with negatively charged nucleic acids. Furthermore, the proportion of such proteins among proteins that do not directly interact with PML is 8% higher than in the full data set.

There are a number of works that discuss the influence of DNA on the biogenesis of PML bodies. For example, in 2004, it was shown [235] that the structural integrity of nuclear PML bodies depends not only on PTMs of PML, but also on the integrity and state of chromatin condensation. In cells not treated with transcription inhibitors, PML bodies are surrounded by chromatin, which explains the stability of their position; direct physical contact between the protein core and chromatin fibers predominate. The inhibition of transcription causes a reduction in these contacts due to chromatin condensation.

PML bodies have been shown to degrade in response to changes in chromatin integrity (demonstrated by introducing exogenous nuclease DNase I into cells) [235]. A 2009 study showed that PML bodies are surrounded by decondensed and condensed chromatin [236]. It has been suggested that PML regulates the incorporation of H3.3 between loose and compact chromatin compartments [237]. In doing so, PML regulates the routing of H3.3 to chromatin by diverting the H3.3 pool and limiting its deposition in chromatin. In addition, there is evidence that PML bodies are involved not only in gene silencing, but also as a potential regulator of epigenetic conditions [237].

## 4. Conclusions

In some of our previous works, we have already raised the question of the need to change existing ideas about the formation and biogenesis of PML bodies [16,19]. In particular, our analysis of the amino acid sequences of PML isoforms showed that the variable C-terminal domains of almost all isoforms of this protein have properties that are characteristic of protein sequences that are potentially capable of phase separation due to electrostatic interactions.

The results obtained indicate a high content of disordered proteins in the composition of PML bodies that are potentially capable of spontaneous separation into phases, and, therefore, allow us to raise the question of changing ideas about the role of nonspecific protein–protein interactions and, possibly, protein–DNA interactions in the formation and biogenesis of PML bodies.

## Figures and Tables

**Figure 1 biomolecules-13-01805-f001:**
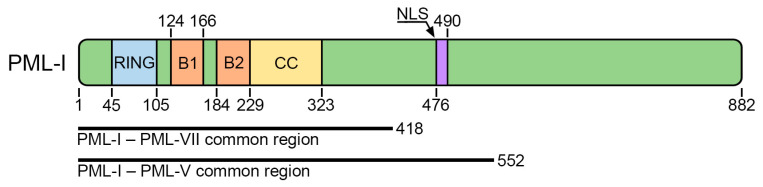
PML protein domain organization scheme. All PML isoforms have the same N-terminal part that contains RING (R), B-Box1 (B1), B-Box2 (B2), and coiled-coil (CC) domains. The majority of PML isoforms contain a nuclear localization signal (NLS).

**Figure 2 biomolecules-13-01805-f002:**
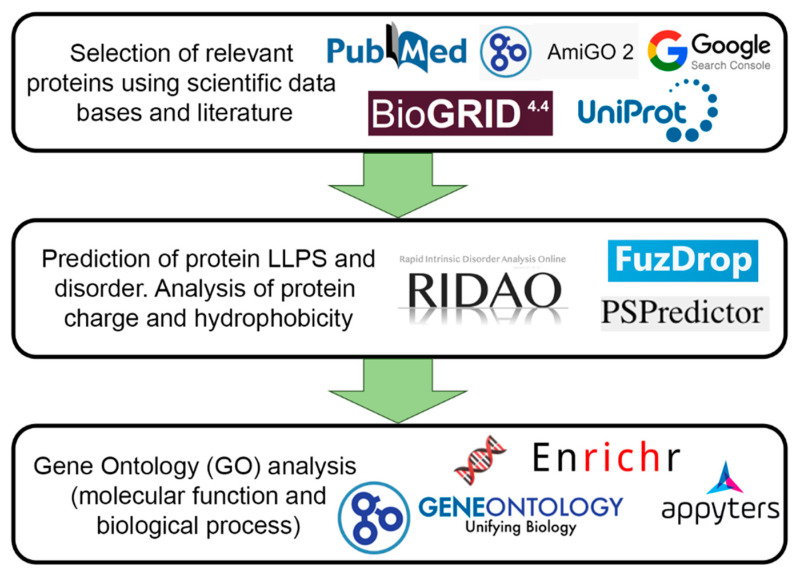
Scheme illustrating the design of our study.

**Figure 3 biomolecules-13-01805-f003:**
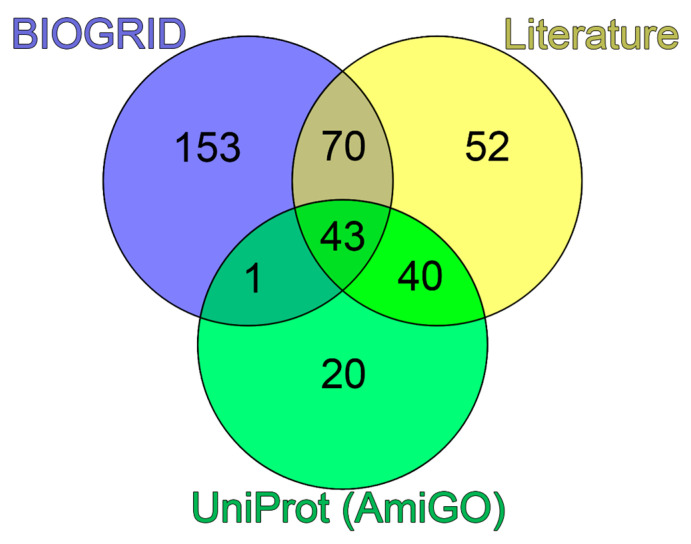
Venn diagram of three sets of proteins potentially included in PML bodies. Blue circle represents the set based on the analysis of the PML interactome in the BIOGRID database. Green circle represents the set according to the analysis of the GO:0016605 and SL-0465 databases. Yellow circle represents the set collected during the analysis of literary sources.

**Figure 4 biomolecules-13-01805-f004:**
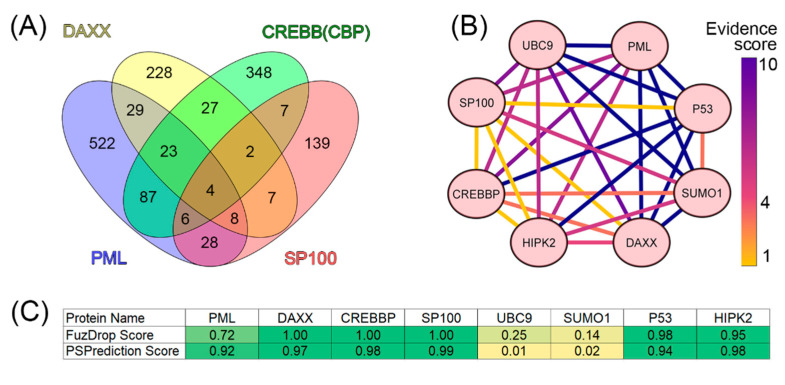
Analysis of the interactome of PML bodies’ scaffold proteins. (**A**) Venn diagram represents the intersection of the interactomes of four PML bodies’ scaffold proteins (PML, DAXX, SP100, and CREBB(CBP)). Data were obtained using BioGRID v4.4. (**B**) Interaction map for eight potential PML body envelope proteins based on the statistical representation in the BIOGRID (evidence score) database. Yellow shows results with weak statistics (data from one experiment), blue shows data with strong statistics (>10 experiments). The heat map was constructed using CytoScape [221]. (**C**) Table of results for predicting the propensity of detected proteins to undergo spontaneous phase separation using FuzDrop and PSPrediction scores.

**Figure 5 biomolecules-13-01805-f005:**
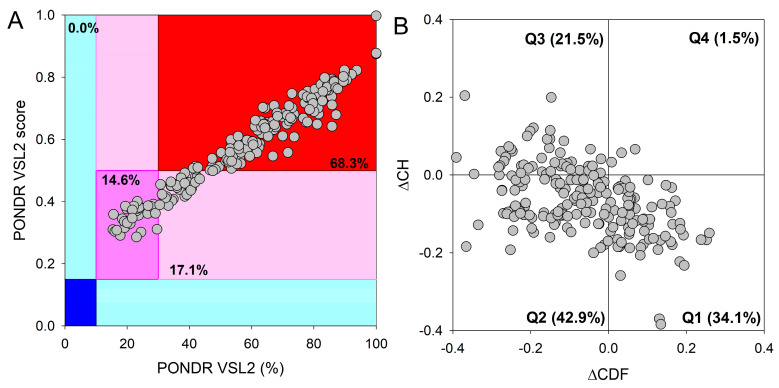
Analysis of global protein disorder within the PML-body proteome. (**A**) The output of PONDR^®^ VSL2, where the PONDR^®^ VSL2 score denotes the mean disorder score (MDS) for a given protein. This score is derived by summing up the per-residue disorder scores and dividing this sum by the number of residues in a query sequence. The PONDR^®^ VSL2 (%) corresponds to the percentage of predicted intrinsically disordered residues (PPIDR), indicating the proportion of residues with disorder scores above 0.5. Color-coded blocks depict distinct regions within the MDS-PPIDR phase space based on the degree of protein (dis)order: mostly disordered proteins are found in the red colored block, moderately disordered proteins are within the pink and light-pink blocks, and predominantly ordered proteins are positioned in the blue and light-blue blocks. Dark-colored background areas (blue, pink, and red) indicate concordance between the two parameters; i.e., MDS and PPIDR, while light-blue and light-pink colors signify areas where only one criterion aligns. The delineation of colored regions is contingent upon the arbitrary yet accepted cutoffs for PPDR (*x*-axis) and MDS (*y*-axis). (**B**) The cumulative distribution function and charge–hydropathy plot (ΔCDF-ΔCH) for the nuclear PML-body proteome comprising 205 proteins. The Y-coordinate is determined by the distance of each protein from the boundary in the CH plot, while the X-coordinate is calculated as the average distance of the protein’s CDF curve from the CDF boundary. The protein classification is based on the quadrant in which it is positioned. In Quadrant 1 (Q1), 70 proteins (34.1%) are projected to be ordered according to CDF and exhibit compact characteristics in the CH plot. In Quadrant 2 (Q2), 88 proteins (42.9%) are anticipated to be ordered/compact based on the CH plot but display disorder as per the CDF plot. Quadrant 3 (Q3) encompasses 44 proteins (21.5%) predicted to be disordered based on the CH plot and CDF. Quadrant 4 (Q4) includes 3 proteins (1.5%) anticipated to be disordered according to the CH plot but ordered according to the CDF analysis.

**Figure 6 biomolecules-13-01805-f006:**
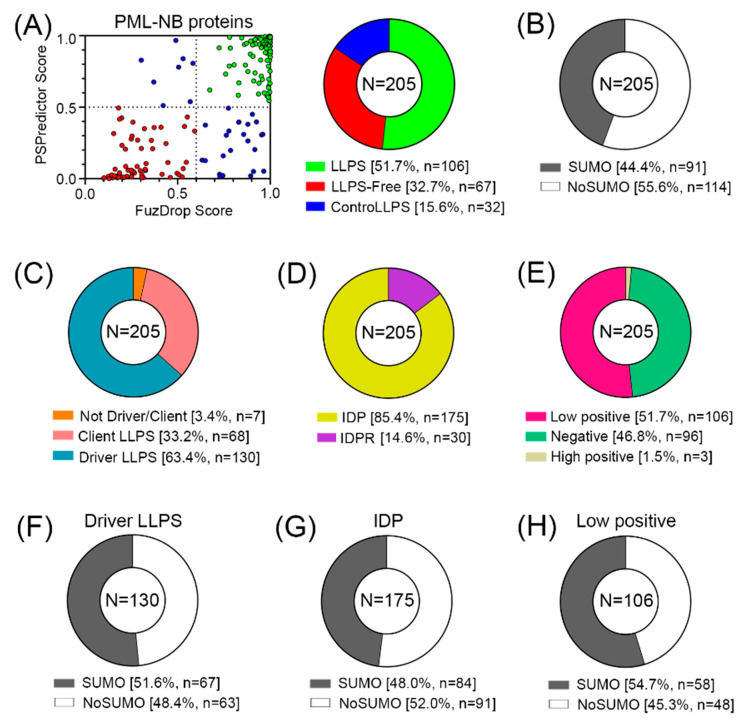
Proteome analysis of PML bodies. (**A**) Correlation of the results of two predictors (FuzDrop and PSPredictor) and the results in the form of a pie diagram. (**B**) Proportion of proteins potentially capable (SUMO) and incapable (NoSUMO) of SUMOylation in the analyzed set of proteins. (**C**) Share of LLPS drivers/clients. (**D**) Proportion of disordered proteins (IDPs) and proteins with intrinsically disordered protein regions (IDPRs). (**E**) Proportion of proteins potentially carrying a negative charge, a weak positive charge, and a positive charge. Proportion of proteins potentially capable (SUMO) and incapable (NoSUMO) of SUMOylation for LLPS (**F**), IDP (**G**) and weakly positive (**H**) drivers.

**Figure 7 biomolecules-13-01805-f007:**
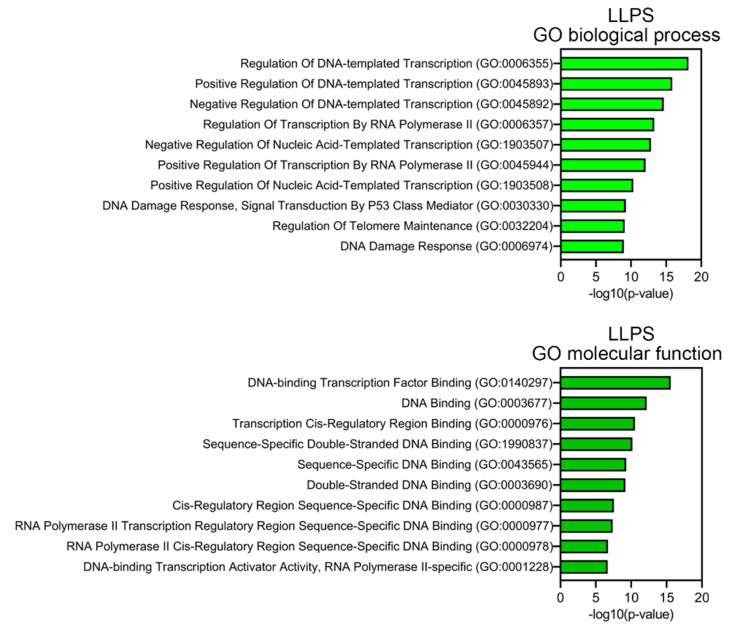
Result of GO analysis for proteins potentially prone to LLPS. Biological process and molecular functions are shown.

**Table 1 biomolecules-13-01805-t001:** Comparative characteristics of the scaffold proteins of PML bodies (except PML protein). DPRs are droplet-promoting regions, AHSs are aggregation hot spots.

Protein Name (Gene Name)	UniProt ID	PER (VSL2B) Disorder	LLPS	DPR Number, Length aa	AHS Number, Length aa	Role in LLPS
Cellular tumor antigen p53 (P53)	P04637	IDP	Yes	4.2	7.7	Driver
Nuclear autoantigen Sp-100 (SP100)	P23497	IDP	Yes	7.5	16.1	Driver
Homeodomain-interacting protein kinase 2 (HIPK2)	Q9H2X6	IDP	Yes	6.3	8.9	Driver
CREB-binding protein (CREBB)	Q92793	IDP	Yes	12.2	30.3	Driver
Death domain-associated protein 6 (DAXX)	Q9UER7	IDP	Yes	6.5	14.1	Driver
Small ubiquitin-related modifier 1 (SUMO1)	P63165	IDP	No	1.2	1.6	Client
SUMO-conjugating enzyme UBC9 (UBE2I)	P63279	Partial IDP	No	0	0	No LLPS

**Table 2 biomolecules-13-01805-t002:** Comparative characteristics of the main nuclear PML isoforms. DPRs are droplet-promoting regions, AHSs are aggregation hot spots.

PML Isoform	UniProt ID	PER (VSL2B) Disorder	LLPS	DPR Number, Length aa	AHS Number, Length aa	Role in LLPS
PML-I	P29590-1	IDP	Yes	5.3	12.9	Driver
PML-II	P29590-8	IDP	Yes	5.4	12.1	Driver
PML-III	P29590-9	IDP	Yes	5.3	8.7	Driver
PML-IV	P29590-5	IDP	Yes	4.2	9.8	Driver
PML-V	P29590-2	IDP	Yes	4.2	7.8	Driver
PML-VI	P29590-4	IDP	Yes	4.2	7.5	Driver

## Data Availability

Data are contained within the article or Appendix A.

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
