# Peer review of "On the Prevalence and Roles of Proteins Undergoing Liquid–Liquid Phase Separation in the Biogenesis of PML-Bodies"

_biomolecules, 2023, doi:10.3390/biom13121805_

Round 1

Reviewer 1 Report

Comments and Suggestions for Authors

A fine piece of work that extends to PMLs the systematic investigation about the biological meanings of protein intrinsic disorder. A paper that is informative, well written and interesting, Nothing of substantial to add. Just one question: Why not to quote, in the introduction, the nice review

Antifeeva, I.A., Fonin, A.V., Fefilova, A.S. et al. Liquid–liquid phase separation as an organizing principle of intracellular space: overview of the evolution of the cell compartmentalization concept. Cell. Mol. Life Sci. 79, 251 (2022). https://doi.org/10.1007/s00018-022-04276-4

And a minor point of style: at the beginning of the introduction (line 36) why to make reference to the "LLPS revolution that occurred in the 2020s". Revolution is a term more apt to political activism than to regular scientific papers. Of course, the whole of us are aware of the classic essay " The Structure of Scientific Revolutions" by T H Kuhn. But frankly speaking there are not scientific revolutions every decade or so. Am I wrong?

Author Response

Reply

We are very grateful to Reviewer 1 for nice comments and high rating of our work.

Of course, we added to references our previous paper.

And a minor point of style: at the beginning of the introduction (line 36) why to make reference to the "LLPS revolution that occurred in the 2020s". Revolution is a term more apt to political activism than to regular scientific papers. Of course, the whole of us are aware of the classic essay " The Structure of Scientific Revolutions" by T H Kuhn. But frankly speaking there are not scientific revolutions every decade or so. Am I wrong?

Reply

Indeed, we have gone too far. We corrected this phrase.

However, I stand assured that Brangwynne and Hyman are good candidates for Nobel Prize in the nearest future

Reviewer 2 Report

Comments and Suggestions for Authors

The authors identify a set of protein involved in the formation of promyelocytic leukemia nuclear bodies (PML bodies) and perform a bioinformatic analysis on the identified set. The authors find that roughly half of the proteins in the set are predicted to promote liquid-liquid phase separation (LLPS) and that half of the proteins in the set  do not contain SUMO binding sites. These findings lead the authors to suggest that non-specific interactions play a role significantly larger than previously thought in PML bodies biogenesis.

The results presented by the authors are in my opinion interesting ones. At the same time I have several doubts on the methodology used and on how they are presented.

- Which are the proteins considered in the analysis and why? It is not entirely clear in the first place  which is the set of 205 proteins used in the analysis. One can obviously guess they are the members of the yellow set in Fig. 1, but this is never explicitly stated. Actually, the sentence at the beginning of section 2.3 is possibly misleading, besides being incomplete. Also, Fig. 7 and the short description in section 31. are misleading, since Biogrid and Uniprot (referred to in the upper box of Fig. 7) do not eventually enter in the selection of the 205 proteins analyzed for LLPS and intrinsic disorder. Moreover, the authors state at the end of section 2.1 that 52 of 205 proteins capable of being included in PML bodies do not interact directly with the PML protein; could this just be due to the lack of experimental information in the Biogrid or Uniprot data? Conversely, how probable is it that the 153 + 1 + 20 proteins connected to PML by Biogrid or/and Uniprot data are yet to be found localized in PML bodies? Also, according to Fig.1, there are 104 proteins in the green circle, not 105, as stated instead at line 111. Finally, the connection between the proteins analyzed in section 2.2 (both the 4 scaffold proteins and the 4 proteins belonging to all of their interactomes) and the set of 205 proteins analyzed subsequently is not clear. I guess that all 8 proteins do belong to the set of 205 but this should be stated explicitly. I do not then understand the relevance of the last sentence in section 2.2 (identifying 4 proteins as PML body scaffolding proteins) in view of the subsequent analysis.

- Is anything known about the relative abundance of different proteins involved in PML body formation? All the results in the manuscript implicitly assume that each of the 205 considered proteins is equally abundant. Could the authors comment on this?

- I do not see the point of the data shown in Figure 4. Those correlations per se look trivial to me. Would they be different if computed, say, in the human proteome? If so, what could we learn from those differences? A brief mention in Materials and Methods of what AHS are should anyway be given.

- I would actually suggest the authors to add a figure with a representation of all the domains of the PML protein, to help readers to follow the discussion at lines 69-73.

- I have an issue with the relationship between the PPIDR and the PER(VSL2B) indicators. There are hints throughout the manuscript that they really are the same indicator; e.g. caption of Fig. 3 stating that PPIDR is the percentage of residues with disorder scores above threshold, which is how PER(VSL2B) is defined in section 3.2 at lines 351-353. The caption of Fig. 4 seems to underscore the same point, relating the indicator referred to in the second row/column of Fig. 4 to the one used in the x-axis of Fig. 3A.  On the other hand, they seem to be introduced in section 3.2 as different indicators, most importantly with different ranges used for the classification of intrinsic disorder content: 50/85 for PER(VSL2B) and 10/30 for PPIDR. The latter is used on the x-axis of Fig. 3A, but then I do not understand the data in Fig. 5D, which would be coherent with  PER(VSL2B)=PPIDR but using a 10/30 threshold for classifying proteins as IDPRs/IDPs.

- The ALT acronym is never defined (line  64)

- Line 207: moonlighting proteins are mentioned only here. Are all 205 proteins in the analyzed set assumed to be moonlighting? On which basis?

- Lines 252-290: mentioning the different Fig. 5 panels in the appropriate sentences would greatly help the reader.

- How the presence of SUMO binding sites is detected (Fig. 5 data) should be mentioned in Materials and Methods

- the reported charge is the average charge (section 3.4); is hydrophobicity as well computed as the average one?

Author Response

Reply

We are grateful to Reviewer 2 for carefully reading our work and providing valuable comments.

We have corrected the text, taking into account all the comments made by the Reviewer 2 in the new version.

- Which are the proteins considered in the analysis and why? It is not entirely clear in the first place  which is the set of 205 proteins used in the analysis. One can obviously guess they are the members of the yellow set in Fig. 1, but this is never explicitly stated. Actually, the sentence at the box of Fig. 7) do not eventually enter in the selection of the 205 proteins analyzed for LLPS and intrinsic disorder. Moreover, the authors state at the end of section 2.1 that 52 of 205 proteins capable of being included in PML bodies do not interact directly with the PML protein; could this just be due to the lack of experimental information in the Biogrid or Uniprot data? Conversely, how probable is it that the 153 + 1 + 20 proteins connected to PML by Biogrid or/and Uniprot data are yet to be found localized in PML bodies? Also, according to Fig.1, there are 104 proteins in the green circle, not 105, as stated instead at line 111. Finally, the connection between the proteins analyzed in section 2.2 (both the 4 scaffold proteins and the 4 proteins belonging to all of their interactomes) and the set of 205 proteins analyzed subsequently is not clear. I guess that all 8 proteins do belong to the set of 205 but this should be stated explicitly. I do not then understand the relevance of the last sentence in section 2.2 (identifying 4 proteins as PML body scaffolding proteins) in view of the subsequent analysis.

Reply

Thanks for your valuable comment. We rewrote this part of the text, made it more logical in our opinion. We attempted to combine known data on the PML-bodies proteome into one array. Since PML protein is localized in the cell not only in PML-bodies and has a huge number of partners, not all PML-binding proteins are included in the proteome of these membrane-less organelles. Therefore, we compared data on the PML interactome obtained from bioinformatics databases with literature data on the localization of proteins in PML bodies, confirmed by electron and fluorescence microscopy (Table S1). For further analysis, we selected 205 proteins, 52 of which do not appear in the Biogrid and Uniprot databases. We understand that proteomic data, especially for dynamic multicomponent structures such as PML-bodies, may be incomplete. We adopted the text to figure 1. In section 2.2 we tried to show that 4  proteins can be classified as scaffold proteins of PML bodies. We rewrote the part of section 2.2. The main idea of this chapter is that the scaffold proteins of PML-bodies, as well as other proteins in their proteome, are prone to spontaneous LLPS.

- Is anything known about the relative abundance of different proteins involved in PML body formation? All the results in the manuscript implicitly assume that each of the 205 considered proteins is equally abundant. Could the authors comment on this?

Reply

The biogenesis of PML-bodies involves both scaffold proteins of PML bodies (constantly present in PML-bodies) and client proteins (temporarily localized in PML-bodies). We rewrote section 2.2 to emphasize this point.

- I do not see the point of the data shown in Figure 4. Those correlations per se look trivial to me. Would they be different if computed, say, in the human proteome? If so, what could we learn from those differences? A brief mention in Materials and Methods of what AHS are should anyway be given.

Reply

We included this figure to show how the analyzed parameters relate to predicting the propensity of any proteins to LLPS. We added information about AHS in Materials and Methods

- I would actually suggest the authors to add a figure with a representation of all the domains of the PML protein, to help readers to follow the discussion at lines 69-73.

Reply

We added this figure

- I have an issue with the relationship between the PPIDR and the PER(VSL2B) indicators. There are hints throughout the manuscript that they really are the same indicator; e.g. caption of Fig. 3 stating that PPIDR is the percentage of residues with disorder scores above threshold, which is how PER(VSL2B) is defined in section 3.2 at lines 351-353. The caption of Fig. 4 seems to underscore the same point, relating the indicator referred to in the second row/column of Fig. 4 to the one used in the x-axis of Fig. 3A.  On the other hand, they seem to be introduced in section 3.2 as different indicators, most importantly with different ranges used for the classification of intrinsic disorder content: 50/85 for PER(VSL2B) and 10/30 for PPIDR. The latter is used on the x-axis of Fig. 3A, but then I do not understand the data in Fig. 5D, which would be coherent with  PER(VSL2B)=PPIDR but using a 10/30 threshold for classifying proteins as IDPRs/IDPs

Reply

Thank you for pointing this out. We corrected this mistake. The PER(VSL2B) indicator was used as a measure of protein structural disorder, which reflects the percentage of residues in the amino acid sequence of a protein for which the prediction of disorder (Disorder Score) is higher than 0.5. Such regions are determined by the algorithm as “probably disordered.”

In a less conservative analysis, the percent of predicted intrinsically disorder residues (PPIDR) was used to classify each protein based on their level of disorder. Generally, a PPIDR value of less than 10% is taken to correspond to a highly ordered protein, PPIDR between 10% and 30% is ascribed to moderately disordered protein, and PPIDR greater than 30% corresponds to a highly disordered protein.

- Line 207: moonlighting proteins are mentioned only here. Are all 205 proteins in the analyzed set assumed to be moonlighting? On which basis?

Reply

The phrase isBased on these classifications, none of the moonlighting proteins was predicted “

- Lines 252-290: mentioning the different Fig. 5 panels in the appropriate sentences would greatly help the reader.

Reply

We added this information

- How the presence of SUMO binding sites is detected (Fig. 5 data) should be mentioned in Materials and Methods

Reply

This information is presented in 3.5 section

- the reported charge is the average charge (section 3.4); is hydrophobicity as well computed as the average one?

Reply

Yes, hydrophobicity computed as the average

Round 2

Reviewer 2 Report

Comments and Suggestions for Authors

The authors addressed successfully most of my concerns. I especially appreciate the new figure 1 and tables 1,2 and the way the selection of the sequence data sets is now explained.

However

- something odd happened with references; a whole bunch of around 200 (I guess) undesired references crept in

- at page 6: Fig3 -> Fig4 (not sure I checked all Figures renumberings)

- I still do not see the point of Figure 5. The authors state in the main text

"Therefore, based on the results of these analyses (the ones reported in Fig. 4), one can conclude that the levels of intrinsic disorder in human proteins found in the PML-bodies proteome are considerably higher than those of entire human proteome, suggesting that intrinsic disorder plays important roles in functions of these proteins and is likely to be related to the biogenesis of this MLO.
This hypothesis was supported by the analysis of the predisposition of human proteins from the PML-bodies proteome to undergo spontaneous LLPS. For this purpose, a combined approach was used, based on the multiparametric bioinformatics analysis of the sequences of the proteins under study (Figure 5)"

Then, after showing Figure 5, the main text goes on referring to the results shown in Figure 6: "The conducted analysis revealed (Figure 6, Table S2) that there are practically no 259 globular proteins in the analyzed data set, ..."

As it stands, there is no discussion in the main text of the data reported in Figure 5. I understand from the caption, that the reported data are obtained by analyzing the 205 sequence identified as connected to PML-bodies, so I do not understand the author reply in this respect, referring to "any protein". The sentence reported above in the main text about the "predisposition to undergo spontaneous LLPS" is my view justified already by the data in figure 6 and the related discussion. If they want to maintain Figure 5, the authors should elaborate more on the reported data.

- please mention in Materials and Methods that you are computing the average hydrophobicity

- moonlighting proteins: I still do not understand what the authors wish to communicate here. Please explain in the main text what you mean by moonlighting proteins.

- in the abstract: SOMYylated -> SUMOylated (I guess)

Comments on the Quality of English Language

The quality of the English language in most of the revised parts needs to be improved.

Author Response

We are grateful to Reviewer 2 for valuable comments. We have tried to correct the text according them.

- something odd happened with references; a whole bunch of around 200 (I guess) undesired references crept in

Reply: These references have been listed in Table S1 to avoid overcrowding the text. But at the request of the editor, we inserted them into the main text

- at page 6: Fig3 -> Fig4 (not sure I checked all Figures renumberings)

Reply: Corrected

- I still do not see the point of Figure 5. The authors state in the main text

"Therefore, based on the results of these analyses (the ones reported in Fig. 4), one can conclude that the levels of intrinsic disorder in human proteins found in the PML-bodies proteome are considerably higher than those of entire human proteome, suggesting that intrinsic disorder plays important roles in functions of these proteins and is likely to be related to the biogenesis of this MLO.
This hypothesis was supported by the analysis of the predisposition of human proteins from the PML-bodies proteome to undergo spontaneous LLPS. For this purpose, a combined approach was used, based on the multiparametric bioinformatics analysis of the sequences of the proteins under study (Figure 5)"

Then, after showing Figure 5, the main text goes on referring to the results shown in Figure 6: "The conducted analysis revealed (Figure 6, Table S2) that there are practically no 259 globular proteins in the analyzed data set, ..."

As it stands, there is no discussion in the main text of the data reported in Figure 5. I understand from the caption, that the reported data are obtained by analyzing the 205 sequence identified as connected to PML-bodies, so I do not understand the author reply in this respect, referring to "any protein". The sentence reported above in the main text about the "predisposition to undergo spontaneous LLPS" is my view justified already by the data in figure 6 and the related discussion. If they want to maintain Figure 5, the authors should elaborate more on the reported data.

Reply: We removed Figure 5 and its consideration from the text.

- please mention in Materials and Methods that you are computing the average hydrophobicity

Reply: Corrected

- moonlighting proteins: I still do not understand what the authors wish to communicate here. Please explain in the main text what you mean by moonlighting proteins.

Reply: Thank you for pointing this out. We changed We removed the term “moonlighting proteins” and changed it to “proteins analyzed in this study”

- in the abstract: SOMYylated -> SUMOylated (I guess)

Reply: Corrected